# Scribal Revelations in Ancient Judaism

Ida Fröhlich

Department of Hebrew Studies, Pázmány Péter Catholic University, 1088 Budapest, Hungary;
frohlich.ida.kinga@gmail.com

**Abstract:** Revelations, visions and their interpretations create in themselves authority. In early Jewish Aramaic tradition, however, this is increased by the role of writing. Enoch receives revelations of the secrets of heaven from heavenly tablets by the Holy Watchers. The Fallen Watchers teach the earthly women magic and sorcery from tablets stolen from the heaven. Scribalism in Second Temple period Judaism and Enoch is becoming more and more researched. As is known, Enoch has a Mesopotamian scholarly tradition behind it, which saw the movement of the celestial bodies as a heavenly writing, the transmission of the will of the gods. Enochic scribes had a good familiarity with the Mesopotamian scribal tradition that took place in the sanctuaries from the Persian period onwards and whose purpose was to record astronomical observations, write diaries, prepare astronomical tables and produce almanacs recording events. Scholarly texts were considered as "secret" or "exclusive" knowledge. The omen list Enūma Anu Enlil, based on a 360-day calendar, was the pinnacle of the scribal tradition and the basis of Mesopotamian astral magic. The Mesopotamian revelatory form in Enoch serves to assert the authority of a calendrical system of its own, the 364-day year and the Holy Watchers and other angelic beings who govern it. The scribal form of revelation is known in Daniel 7 (also in Aramaic), in which the books opened in heaven contain a revelation about the fate of the fourth empire. The book-revelation of cyclic and linear time is present together in the book of Jubilees, whose chronology is based on the 364-day year, and in which Enoch keeps a record of earthly events on heavenly tablets.

**Keywords:** Judaism; revelations

## 1. The Babylonian Exile

The saying that "prophecy died in the exile" is fundamentally true. In 586 BCE, following an unsuccessful rebellion of the Judaean king Zedekiah, the armies of the Neo–Babylonian ruler Nebuchadnezzar captured Jerusalem, the capital of the kingdom of Judah. Tradition says that the Temple was looted and destroyed. A significant part of the population was taken into captivity to Mesopotamia and resettled there. The date 586 BCE became a symbol of the exile. However, several members of the royal family and representatives of the priestly and secular aristocracy were taken into Babylonia as early as 597, in retaliation for an earlier attempt at rebellion. Among them was a young priest named Ezekiel.[1]

The phenomenon indicated by these events was not new. Almost a century and a half earlier, in 722 BCE, the Neo–Assyrian empire, predecessor of the Neo–Babylonian empire, had occupied the northern kingdom Israel and settled thousands of its inhabitants in northern Syria and other parts of the empire.[2] The story of "exile" thus begins with this Northern group, although almost nothing is known about their later history.[3] Judaean diaspora is better documented. Settled in groups, they were able to maintain their institutions and social network.[4] Zecharia knows of "a temple in the land of Shinar", an institution (temple of other) that could have served the needs of the community living there (Zech 5:9–11). Ezra repeatedly mentions a place called Casiphia from which priests were recruited to properly administer temple services in Jerusalem (Ezra 8:15–20).[5]

One of the most important changes caused by captivity was the change in language. However, the language adopted by the exiles was not Akkadian, spoken in Mesopotamia for thousands of years and with a huge cuneiform culture, but Aramaic, a West-Semitic language that by that time had become the vernacular of Mesopotamia.[6] Aramaic was the *lingua franca* in Mesopotamia, becoming multi-ethnic due to natural Aramaic infiltration and imperial resettlement policies from various territories. Later, the Persian rulers made Aramaic the language of administration throughout their empire, where a unified scribe culture emerged from Afghanistan to Egypt. Aramaic, which used alphabetic writing, had several advantages over the cuneiform used in Mesopotamia, which took several years to learn. Although it was possible to use cuneiform at a lower level in everyday life (contracts of merchants, civil documents), scientific and literary texts were taught at higher school levels (Van De Mieroop 2023). Alphabetic scripts such as Aramaic were easier to learn, and the flexibility of the writing made it suitable for use in different types of text and languages. From the beginning, Aramaic was also a mediator of Mesopotamian science and literature for the population of foreign origin. Unfortunately, due to the perishability of the writing medium (papyrus, leather, wax tablets), very little of the Aramaic scriptures have survived. The catalogues of Assubanipal's library in Nineveh refer to scrolls (*magillatu*), which were presumably written in Aramaic, but were destroyed by the library's burning—as opposed to clay tablets, which became even more resistant when burned.[7]

The conquerors always took with them the social and cultural elite at the first stage of conquest. It was members of the cultural elite who met the culture of the conquerors at the top level and who were able to convey this culture to their own group of origin. Bilingual Aramaic scribes were needed in the administration, who carried their own former culture with them and were able to master the local clerical culture. This situation is vividly depicted in the opening chapter of the book of Daniel (although it was written much later than the events): "Then the king commanded his palace master Ashpenaz to bring some of the Israelites of the royal family and of the nobility: young men without physical defect and handsome, versed in every branch of wisdom, endowed with knowledge and insight, and competent to serve in the king's palace; they were to be taught the literature and language of the Chaldeans" (Dan 1:3–4). Members of the higher clerical and priestly classes were certainly familiar with the cuneiform tradition. The encounter with Mesopotamian culture resulted in a significant impact on the literature of the Eastern Jewish diaspora. However, impact never means a takeover, but an adequate response to new cultural challenges in a critical era and a new place of residence.

The exile was an era of disorientation and discontinuity.[8] One natural response to these challenges was to return to the beginning, in the spirit of a renewed theology. The Persian rulers supported the cult of Jerusalem. Two large corpuses were created during the Persian period. One of them is the collection of the earlier prophetic books, which was selected and edited at that time. All prophetic books have survived in exilic revision. The same is true of historiography, in which the existing narrative tradition was expanded and edited in the spirit of a new theology and with new literary knowledge, and the whole was placed within a framework of world history created during the exile (Blenkinsopp 1996, pp. 148–93) (chap. V, Between the Old Order and the New).

Although the Exilic prophets wrote in Hebrew, they seemed to be well versed and appreciative of contemporary events and familiar with cuneiform literature. Prophecy is essentially a public genre, with its orality and symbolic actions fulfilling an important social role. Preexilic prophets manifested themselves publicly on social and political issues, addressing the king and higher social classes. Constant themes were international political issues and social injustice. However, prophets were not representatives of certain social groups, but acted as the "tongue of the balance". They formulated their judgments from a theological point of view, focusing on the covenant, the relationship between the community and the deity who protected it.

Captivity destroyed the former social framework, the social arena for prophetic activity. Prophecies were passed down under pseudonyms. Among the works of the prophets of the

Persian era—Haggai, Zechariah, Malachi, Second and Third Isaiah (Isa 40–55; 56–66), Joel, and Jonah, Ezekiel—there are only two books that have left anything like the portrait of an actual person: Haggai and Zechariah (Grabbe 2004, p. 251). At the same time, however, some still have the traditional prophetic role, the need to legitimize kings. Deutero–Isaiah (Isa 40–55) speaks in the spirit of the Cyrus cylinder when he recognizes Cyrus as the legitimate ruler ordained by Yahweh for his people (Isa 40:45–46). In addition, vision is an increasingly common form in postexilic prophecy, and visions are also carriers for expressing political expectations.[9]

Theological dilemmas of the exile and answers are formulated by Ezekiel, also called "the prophet of exile",[10] who was by no means isolated from either the cuneiform culture or the literature of the diaspora, as it is indicated by his themes and vocabulary.[11] The vision in the first chapter of his book (repeated in chapter 10) reflects on the opinion that God's glory (*kābōd*) has departed Jerusalem when His sanctuary was destroyed.[12] According to ancient Near Eastern understanding, temples were the abode of gods who protected cities and countries. Their destruction meant that the deity left the sanctuary and the community it had protected until then. The location of Ezekiel's vision is emphatically Mesopotamian and seemingly profane: the banks of the "river" Kebar. Ezekiel sees the heaven opening above his head. He sees the cosmos and the forces that govern it, the four "beings" (called cherubs in chapter 10) representing the turning points of the Sun's annual orbit, governed by the divine spirit (*rwḥ*). Above all this, divine glory (*kābōd*) is enthroned, as if in human form, shining in metallic light. The message of the vision is that God is present among those in captivity.

The prophetic spirit works in Ezekiel in several ways. He mentions that "I was filled with spirit, he set me up, and I heard him speak to me" (Ezek 2:1) or that "Then the hand of the Lord was upon me there" (Ezek 3:22). Before his cosmic vision, "the heavens were opened and I witnessed a divine vision" (Ezek 1:1). Another time the spirit (*rwḥ*) lifted him up and brought him to Jerusalem, "to the east gate of the house of the Lord, which faces east" (Ezek 11:1). Ezekiel also relates a strange case of scribal revelation when a hand reaching out from heaven gives him a written scroll and instructs him to eat the scroll. He then conveys the contents of the scroll in the form of prophetic revelation (Ezek 2:9–3:4).

## 2. Revelations in Aramaic

Scribal revelatory traditions also arose in Aramaic-speaking communities of the Jewish diaspora. One of them is Danielic tradition. Daniel is considered a prophet by some traditions, but not by others.[13] Part of the collection surviving under his name was written in Aramaic (Dan 2–7) and others in Hebrew (Daniel 1, 8–12). The core and earliest part of the collection are the Aramaic chapters 2, 4 and 5 of the collection (Koch 1980). Of these, Dan 2 is of decisive importance for later pieces in the collection and also for other works written in the Jewish diaspora. Prophecies appear in these chapters in a new form as interpretations of visions and texts. In Daniel 2, a great statue appears in King Nebuchanezzar's dream vision: "The head of that statue was of fine gold, its chest and arms of silver, its midsection and thighs of bronze, its legs of iron, its feet partly of iron and partly of clay" (Dan 2:32–33). The statue is hit by a stone (according to the version of the LXX: "from a mountain"), which is "hewn . . . by no human hand, and shatters the image" (Dan 2:34). The vision is interpreted by Daniel, an exiled Jew who receives the dream and its interpretation in a dream from God. According to the interpretation the four parts of the giant statue represent four successive "rules" or "kingdoms" (*malku'*), a term that can be interpreted in two ways: as the rule of a king or as the rule of a dynasty for a longer period. The author of the interpretation of the vision (introduced by the term *pišra'* "dechiperment") in the Persian period probably identified the first "rule" with a person, King Nebuchadnezzar—at least this is what the words of identification indicate: "You, o king . . . you are the head of gold" (Dan 2:37–38). In this case, the fourth, divided rule would be identified with that of the last Neo–Babylonian ruler, Nabonidus, and the stone that would destroy the statue would

be Cyrus, the Persian ruler who defeated the Neo–Babylonian empire (whom the Jewish prophet Deutero–Isaiah of the Persian period calls *mašiah*, or anointed one, cf. Isa 45:1–7).[14]

The subject of Dan 5 is the famous scene of the "writing on the wall", the interpretation (*pišra'*) of a written text—originating from God. The four enigmatic words (*menē' menē' teqēl ûparsîn*, according to the Masoretic Text) were deciphered by Daniel/Beltshazzar by reading the Babylonian inscription in Aramaic, giving new meaning to the text. Paronomasia, the above method, was a popular exegetic tool applied already by the Mesopotamian commentaries of the first millennium BCE (Esztári and Vér 2022). This prophecy also speaks of the fall of the Neo–Babylonian Empire and the takeover of Persian rule. The prophecy, therefore, has not lost its political interest.[15]

All interpretations reflect the in-depth proficiency of Aramaic authors in Mesopotamian cuneiform writing and its culture, where dreams were collected and organized into collections.[16] The interpretations discussed the meaning of each dream element by element. Dream specialists could consult the collections and give decipherments to specific dreams. But the Book of Daniel provides this old tradition with a new basis: here, the interpreter is not a Babylonian, but the son of captivity, an exile from Judah, and the basis of his interpretation is not learning, but inspiration. The source is the spirit that is in Daniel and comes from his God.

Intertwined with another visionary prophecy, that of the four kingdoms and the fall of the fourth one, Dan 7 describes a scene of the seat of a heavenly court. The members of the court deliver a judgment over the fourth kingdom, symbolized by a "terrifying and dreadful and exceedingly strong" animal, presumably an elephant (Dan 7:7). The "fourth kingdom" can be identified with the Seleucid empire, the fall of which is expected to the time of its eleventh ruler, Antiochus IV Epiphanes (175–164 BCE).[17]

The session is presided over by an Ancient of Days ('*tyq ymyn*) who sits on his throne and before whom the members of the court stand. He is described in anthropomorphic terms, as an old man with white clothes and white hair (Dan 7:9). The figure is associated with motifs of fire: his throne is made of blazing flames, the wheels of the throne are of glowing embers and a river of fire flows from under the throne (Dan 7:9–10). The members of the court hold a verdict, during which "books (*spryn*) were opened", and a judgment pronounced (Dan 7:10). No adjective can be read about the members of the heavenly court, they only are referred to by their multitude: "A thousand thousands (*'lp 'lpym*) served him, and ten thousand times ten thousand (*rbw rbwn*) stood attending him" (Dan 7:10). The words of the judgment (*dyn'*) are pronounced unanimously by the members of the court, as they read it from the books (*spryn*) opened during the session. Aware of the historical and cultural milieu of the text, it can be assumed that "opening books" means opening sealed scrolls of writing, the language of which, like the description of the night vision itself, was Aramaic. There is no reference to any trial that usually precedes judgments—in this case, a presentation of the misdeeds of Antiochus IV Epiphanes. The judgment declares the end of the fourth kingdom and announces the transfer of rule (*mlkw*) to a new leader.[18] It is not known what was the source of the unanimous verdict handed down by the members of the court, i.e., what was the master copy of the text in the books.[19]

## 3. The Aramaic Enochic Tradition

The theme of fall and punishment in Dan 7 evokes another Aramaic work, the Enochic collection called 1Enoch, whose basic story is about rebellion against God and the punishment of sinners, and which served as a basis and model for historical assessments.[20] The story (1En 6–11), which is the core tradition of the work and is repeatedly quoted and interpreted in later parts of the collection, tells of 200 celestial beings called Watchers, who, seeing the beauty of the daughters of men, had a desire to descend among them and beget children with them. They rebelled against God by leaving their appointed places. The leader of the rebels was Shemihazah, under whom 20 leaders of 10 belonged. The Watchers, who became unclean in their relationships with earthly women, taught the women witchcraft and sorcery. Children born of uneven relationships became destructive

giants whose actions further multiplied the impurity of their fathers. The leaders of the Fallen Watchers, Shemihazah and Asael were punished by the chief leaders of the Holy Watchers, Michael, Sariel, Raphael and Gabriel. Bound and cast into darkness, they await their judgment, which will come at the end of time.[21]

The punishment of the Fallen Angels became a historical model, which also served as a model for describing the fall of the fourth kingdom in Dan 7.[22] The scene of the heavenly court is also described in another place in the Enochic collection (1En 14:15–23 // 4Q530 2 ii 17–18). Other examples of the motif are found in Zechariah's vision (Zech 1:7–11) and in the book of Job, where Job's fate is decided by a body of "sons of God" (Job 2:1–7).[23] The scenes do not represent a coherent tradition, but are independent adaptations of a topic, some motifs of which were freely shaped by the authors.

The earliest manuscripts of an Aramaic Enochic collection were found at Qumran. They may have been written at the end of the third century BCE, probably in the Eastern Jewish diaspora.[24] The fragments represent three groups of texts: the Book of the Watchers (1En 1–36), which also contains the story mentioned above, with a text essentially identical with that of the Ethiopic translation; an Aramaic Astronomical book, the fragments of which show thematic correspondence with the Astronomical book of the Ethiopic translation (1En 72–82); and fragments of a collection not known elsewhere, called the Book of Giants. The content of the Book of Giants is based on the tradition of the Fallen Watchers, and the fragments relate dreams and visions on the fate of the sons of the Fallen Watchers, the Giants. Based on content reasons, it can be assumed that the core tradition of the story of the Fallen Watchers goes back to the Persian era.[25] As it was mentioned, the manuscript tradition is assumed to originate from the eastern diaspora, on the basis on the type of the writing of the earliest manuscripts. The content of the manuscripts reflects a strong Mesopotamian background.[26]

Let us start with the calendar: the Astronomical Book (1En 72–82, contained in the Ethiopic text) is primarily a cosmology, on the basis of which a calendar can be set up. This calendar represents an ideal year of 364 days, consisting of four quarters. The first 2 months of each quarter have 30 days, and the third has 31 days. The last days of the 31-day months correspond to the 4 epagomenal days of the year, the solstices and the equinoxes.[27] The 364-day year is not a real but an ideal solar year that does not take into account lunar phenomena and does not show any irregularities (and, let us add, it also reflects a perfect division of the week, since it consists of exactly 52 weeks). The 4 epagomenal days have a cardinal role in the system of the 364-day year. These days are the cornerstones of the year: they mark the boundaries of the seasons and define the year.

The 364-day ideal calendar known from the Enochic collection is the reworking of the 360-day Mesopotamian ideal calendar, based on the monthly journey of the moon through the constellations in its orbit, "in the path of the moon" (Koch 2007). The system is best represented in the compendium called MUL.APIN that lists 17 or 18 constellations, starting with the Pleiades (MUL.MUL) (Pingree 1998, p. 127; Hunger and Steele 2019). The ideal year of 360 days is divided into 12 equal months of 30 days.[28] It has been shown recently that astronomical fragments of the Aramaic Enochic collection (4Q208, 4Q209) reflect the transformation of the 360-day ideal calendar into a 364-day ideal zodiacal calendar.[29] This calendar, observing the relationship between the sun and the 12 constellations in its orbit, measures the change in the ratio of light and darkness throughout the year.

Mesopotamian thought believed that the gods wrote their will in the movements of the celestial bodies, and that the apparent movement of the stars in the night sky and their relative positions were a kind of "celestial writing" that could be read. It was for this reason that individual phenomena were carefully observed and described in order to draw conclusions about the will of the gods and to plan human activities. Conclusions based on long observations were recorded in omen lists.[30] This was summarized in the collection *Enūma Anu Enlil*, a collection of interpretations of *omina* based on the calendar of MUL.APIN. *Enūma Anu Enlil* represented the pinnacle of the written tradition and the basis of Mesopotamian astral magic, a concept that became dominant in Mesopotamia by

the middle of the first millennium BCE. Astral magic served as the basis of all sciences, above all medicine (*melothesia*) (Geller 2010, p. 163; 2014). In its time, the system of astral magic represented a scientific revolution and a change in the world view (Heeßel 2005, pp. 20–22).

In Mesopotamia, scientific texts were considered to contain "secret" or "exclusive" knowledge (Akkadian *pirištu*),[31] and these were referred to as "tablets of the secrets of heaven". Scholars versed in this literature of omen interpretation were called "the scribes of *Enūma Anu Enlil*" (*ṭupšar Enūma Anu Enlil*) (Rochberg 2010, pp. 237–56 (Scribes and Scholars: The Ṭupsar Enuma Anu Enlil)). The center of Babylonian scientific scribal culture in the Neo–Babylonian era was the royal court; later on, in the Persian and Hellenistic periods, centers were increasingly the temples. The task of the scribes included creating almanacs recording events, writing diaries and compiling astronomical tables (ephemerides), which were the interpretation of events.

Based on the content of the texts 4Q208 and 4Q209, it can be assumed that the authors of the Aramaic Enochic collection, in addition to the acquaintance with the 360-day ideal calendar, were not unfamiliar with the Mesopotamian ideas about the heavenly bodies and their functioning. They presumably also were familiar with a written tradition of calendar-based astral magic. Scholars have previously drawn attention to the fact that the teachings of the Watchers regarding the interpretation of celestial signs show a familiarity with the tradition of the *Enūma Anu Enlil* collection of omens (Kvanvig 2011, p. 502). Mesopotamian astral magic was based on the idea that celestial bodies in the night sky affected the earthly world. As stated in the Old Babylonian prayer Gods of the Night, at night, when both men and gods sleep, only the stars are awake and watch over the earthly world. They make decisions instead of the gods and can influence people and the material world with their irradiation.

## 4. Mesopotamian Astral Magic and Jewish Exiles: The Story of the Watchers

It therefore seems obvious to examine the story of the fall of the Watchers in the light of the concepts of astral magic. The majority of the chiefs of the Fallen Watchers bear names that are related to those of celestial bodies and natural phenomena.[32] Their common epithet, "sons of heaven" (*bny šmy'*), also shows them as representatives of the heavenly world. The key sentence for their story lays in Enoch's words to the Watchers in the introductory part of the collection: "But you have not stood firm nor acted according to his commandments; but you have turned aside" (1En 5:4), that is, Enoch considers the operation of the Watchers to be irregular and imperfect; consequently, the calendar based on their operation is also considered irregular and inaccurate.

The story that follows (1En 6–11) is about changes and disturbance in the cosmic system. The Watchers follow their own desire and will when they leave their celestial places to go down to earthly women. They become impure through their relations with the women. The women are taught sorcery and witchcraft. Children born of their relationships are destructive giants that multiply the impurities on Earth.

The personal qualities of the Watchers are similar to those that astral magic ascribes to celestial bodies (Reiner 1995, pp. 1–14 (Introduction), pp. 15–24 (The Role of the Stars)). The name Watcher ('*yr*) itself can refer to the stars of the night sky that guard the earthly world.[33] The Watchers appear in military order and hierarchy, similar to the "host of the heaven", the army of the stars of the nocturnal heaven in ancient beliefs.[34] Similarly to the celestial bodies of astral magic, the Watchers are able to feel a desire that drives them to descend on females. They make their decisions of their own free will. Their rebellion means leaving their places, not armed struggle. They are capable of procreation, similar to the malevolent stars that, as it were, procreate disease in the human body.[35]

The author(s) of the story condemns astral magic. The actions of the Watchers unleash destructive forces hostile to humans, resulting in anomalies and impurities. The story ends with a double punishment: the binding of the leaders of the Watchers and the punishment of the Flood in which the Giants themselves perish. Harmful spirits are released from their

corpses, which continue to act in the world and cause diseases in people (1En 15). The story often recalled in the Qumran texts is an etiological myth, about the origin of physical or natural evil. The story is subversive in nature and serves polemic purposes against Mesopotamian astral magic.

The cosmological counter-tradition of astral magic was created in the Astronomical Book of Enoch (1En 72–82), which contains the calendar of an ideal year of 364 days and its mythical cosmology. In this system, four Holy Watchers stand over the four epagomenal days of the year who oversee all celestial phenomena (1En 82:11). The 12 "leaders of thousands" watch over the 12 months and the 360 days (1En 82:11). The orderly operation of the entire celestial system is overseen by celestial beings, "This is the law of the stars which set in their places, at their times, at their set times, and in their months. These are the names of those who lead them, who keep watch so they enter at their times, who lead them in their places, in their orders, in their times, in their months, in their jurisdictions, and in their positions" (1En 82:9–10).[36] Celestial beings have no emotions and no will of their own. All their activities are aimed at the perfect service of the operation of the celestial system directed by God. The Enochic Astronomical book is a perfect example of apocalyptic revelations, the subject of which is not only history, but the regularities of the workings of the natural world.[37]

### 5. Communication between Heaven and Earth in Aramaic Enoch

How does Enoch learn about all this? The reports in the Astronomical Book mention different ways of receiving knowledge.[38] Some of them claim that these things were shown to Enoch by "the holy angel" Uriel, one of the main Watchers, who controls the movement of all celestial bodies and oversees the entire system (1En 72:1; 82:7). This mode of communication presupposes some kind of personal contact, a celestial journey, during which Enoch traversed the cosmos. He also became personally close to the angel, who, as Enoch says, "breathed on me" (1En 82:7). Uriel, being a spiritual being, conveys the divine spirit (*rwḥ'*) to Enoch with his breath, thus authenticating Enoch's knowledge gained during his celestial journey. Uriel shows Enoch the calendar system and its chronology: "The account about it is true and its calculation is precisely recorded because the luminaries and the months, the festivals, the years, and the days he showed me, and Uriel, to whom the Lord of the entire creation gave orders for me regarding the host of heaven" (1En 82:7). The plausibility of the idea of a celestial journey is supported by another account of Enoch's celestial journey in chapters 17–36 of the Book of Watchers, during which heavenly and earthly landscapes were shown to Enoch.

According to one account, during the presentation held by Uriel, Enoch takes notes that relate to the position of celestial bodies and the calendar: "All this Uriel the holy angel who is the leader of them (i.e., the celestial bodies) all showed me. Their positions I wrote down as he showed me and I wrote down their months as they were and the appearance of their light until fifteen days were completed" (1En 74:2).

According to another tradition, also found in the Astronomical Book, Enoch received revelation about the working of the celestial world from celestial tablets. Enoch was able to read and understand the tablets: "He (i.e., Uriel) said to me: 'Enoch, look at these heavenly tablets, read what is written on them, and understand each and every item. I looked at everything on the heavenly tablets, read everything that was written, and understood everything'" (1En 81:1–2).[39]

Tablets are mentioned in several Enochic and non-Enochic Aramaic texts, in various contexts. A fragment of the Book of Giants contains a conversation about the fate of the Giants. The text mentions two tablets (*lwḥ'*) that were given to someone (probably Enoch): "a copy of the s[ec]ond tablet of the let[ter] in a do[cu]ment of the hand of Enoch, the scribe of interpretation" (*pršgn lwḥ' tn[y]n' dy 'y[grt'] bktb yd ḥnwk spr prš'* [...]) (4Q203 (4QEnGiants-a ar) 8:3–4). So, Enoch makes his own handwritten copy of a letter written on a tablet. It is not known in what language or script the copy was made, Akkadian cuneiform on a tablet or, more likely, Aramaic alphabetic script on papyrus or parchment.

Enoch reads this copy before the Fallen Watchers. The text is characterized by the use of terms of chancellery language ("copy", "letter"). [40] The tablet contains a judgment, which is based on the sins of the Fallen Watchers listed on the same place, primarily on the basis of their fornication (4Q203 (4QEnGiants-a ar) 8:3–15).

Another fragment of the Book of Giants speaks of a tablet (*lwḥ'*) emerging from the waters of the Flood 2Q26 (2QEnGiants ar) 1:1–4. Interpreted in the context of tradition, this tablet presumably also contains the doom of the Watchers.

Tablets are mentioned in another part of the Aramaic Enoch, the story of Noah's birth (1 En 106). Enoch refers to himself here as one "who knows the mysteries (*rz'*) of the Holy One, which God himself has shown him and made known to him. He mentions the tablets as having read them personally: "and I myself read them on the heavenly tablets" (1En 106:19 // 4Q204 (4QEn-c ar) 5ii:26–27). Enoch receives here a revelation about the eras of human history, the series of periods each classified as "good" or "bad", until "evil and wickedness will end, and violence will cease from the earth, and good things will come upon the earth to them" (1En 107:1 // 4Q204 (4QEn^c ar) 5 ii. 27–28). This is, therefore, about the whole of human history, in whose examples the characters of epochs considered "bad" are the embodiment of the transgressions of the Fallen Watcher. They are rebels against God's laws, but their activity is finite.

Heavenly tablets with content similar to the former are mentioned in the fragmentary text of 4Q180–181, which contains a list of examples of good and bad periods of history (4Q180 3). From the fragmentary text, only the mention of the initial "bad" period marked by the fall of the Watchers has survived.

The Astronomical Book also mentions another form of written celestial revelation, the book. The revelations given in books always contain revelation about the course of time: "The entire book about them, as it is, he showed me and how every year of the world will be forever, until a new creation lasting forever is made" (1En 72:1).[41] "I read the book, all the human actions and of all humans who will be on the earth for the generations of the world" (1En 81:2). These books are about history and the judgment of certain individuals.[42] Later parts of the collection mention books with similar content that are in heaven (lists, punishment of sinners).[43] However, books, unlike tablets, are no longer "master copies", but serve practical purposes to judge sinners. They are usually associated with Enoch, who handles the books in heaven (and perhaps records human deeds so that his lists can then be used to testify at judgment).

The transmission of tradition also takes place through different channels. Enoch verbally conveys celestial messages to human audiences on numerous occasions on various subjects, such as the orderly functioning of nature—which means not only the regular functioning of celestial bodies, but also the weather and vegetation, also presenting a catalog of evergreens (1En 1–5). He also verbally tells the story of the Watchers and a list of their names (1En 6–11). After a celestial vision, he verbally informs the Fallen Watchers of the contents of the vision, namely that they have lost their former heavenly status. In his words, he refers to a determined judgment (*dyn gzyr*) passed on them ordering them to "to bind you until all days of eternity" (4Q204 (4QEn^c ar) 1 vi.14–15). However, the "decided judgment" to which it refers indicates a written document.

Enoch, having returned from his celestial journey, passes on the knowledge he gained there to his son Metushelah. One way to do this is to show him the phenomena in the same way that they were shown to him.[44] "Now my son I have shown you everything, and the law of all the stars of the sky is completed" (1En 79:1). Usually, however, he also describes revealed things: "Now my son Methuselah, I am telling you all these things and am writing (them) down" (1En 82:1). What Enoch describes is not a simple transfer of information. His books always contain special knowledge that grants protection and special status to the holders of this knowledge ("they sleep no more"). The knowledge must be passed down to the descendants: "Wisdom I have given to you and to your children and to those who will be your children so that they may give this wisdom which is beyond their thought to their children for the generations. Those who understand will not sleep and will listen with

their ear to learn this wisdom. It will be more pleasing to them than fine food to those who eat" (1En 82:2–3).[45] Another statement claims that after his journey to heaven, Enoch was given 1 year to write down the teachings revealed to him, and his final "taking away" only happened after this time. Enoch is brought back to Earth by seven holy angels who say to him: "We will leave you with your son for one year until (you receive) another order, to teach your children and write for them, and you will testify to all your children; in the second year they will take you from them" (1En 81:5–6).

When it comes to transmitting knowledge to humans, there is never mention of "tablets" (*lwḥ'*), only verbal communication and books. Enoch represents the type of Aramaic scribe who is able to read cuneiform tablets and convey their contents in book scrolls in lettering.[46] Incidentally, the Enochic collection itself contains Enoch's written and collected revelations. In the same way, he acts as an Aramaic scribe when he conveys requests from Earth to heaven.

## 6. The Scribe Enoch

Communication between the celestial and terrestrial worlds is also the subject of sections outside the Astronomical Book. In these passages, Enoch appears not only as a communicator of heavenly secrets, but also an intermediary, a mediator of earthly requests to the heavenly world. He writes and then reads the Watchers' petition, which is called the "book of remembrance" (*spr dkrwn*), and which is a kind of memorandum (4Q204 (4QEnᶜ ar) 1 vi.3). The celestial world in this pericope appears as a royal palace: "[I lifted up] my eyelids up to the gates of the pa[lace of Heaven (*ltrʿy hykl šmyʾ*)] and I saw visions …" (4Q204 (4QEnᶜ ar) 1 vi.3–4).[47] As a result of their fall, the Fallen Watchers lost the ability to write, which is why Enoch writes their petition instead (4Q204 (4QEnᶜ ar) 1 vi.12). According to terminology (*spr*), the medium that conveys the writing containing the request may be a scroll of parchment or leather, and its language is Aramaic.

The figure of the scribe Enoch, the tablets he is able to read and transmit their contents in copies called "books", as well as the terms associated with the Watchers' petition (copy, letter) evoke a chancellery setting. The simultaneous use of Akkadian cuneiform and Aramaic alphabetic script begins in the Neo–Assyrian period in both core areas of Mesopotamia, Assyria and Babylonia, and continues into the Neo–Babylonian period (Van De Mieroop 2023, pp. 161–69). The official form of communication is cuneiform and the Akkadian language. However, Aramaic documents also appear, although the surviving documents indicate that the number of the latter is small compared to the cuneiform tablets (a few hundred compared to several thousand or tens of thousands of items). Bilingual texts are also included in administrative and legal texts; some cuneiform texts are accompanied by a brief Aramaic summary, probably written by the same scribe. This practice evokes the imagery on the walls of Assyrian palaces, where a cuneiform scribe holding a clay tablet appears next to the Aramaic scribe holding a pen and scroll, who immediately translates the dictated text.

It is not known how many and what works were written in Aramaic, nor whether there were Aramaic translations of the cuneiform works. The everyday language of the deportees, including that of the Jews, was Aramaic, and they communicated with their surroundings in this language. In some circles, Hebrew remained the literary language, which later became a sacred language. Ezekiel and the surviving authors under his name wrote in Hebrew. At the same time, however, they all know and quote cuneiform traditions.[48] The first known Aramaic literary work is the Ahiqar novel, whose author (or compiler) is believed to be an Aramaic scribe, "Aba-enlil-dari, whom the Arameans call Ahiqar". He is mentioned as a contemporary of Esarhaddon (681–669 BCE), and the work associated with his name is also referred to.[49] This work is known from a copy read by members of a Jewish military colony in Elephantine in the middle of the fifth century BCE.[50] The story is about a wise, honest and innocently accused official whose threatening fate turns good by chance as a reward for a previous good deed. The collection of wise sayings attached to the story emphasizes adherence to moral values as the only guarantee of official and

social success. The story and parables about adherence to moral values and their divine reward could serve as an example for all outcasts. The Ahiqar novel had a huge impact on Judeo–Aramaic literature. The Aramaic narratives of court officials in the book of Daniel (chapters 3 and 6) were inspired by this narrative. The Book of Tobit (4th-2nd century BCE), mentions Ahiqar by name as Tobit's nephew and weaves his story into his own narrative, which is also a career story.

The majority of surviving Aramaic literature is Judeo–Aramaic.[51] The most extensive among them is the Enochic collection, the oldest pieces of which were probably written in the Jewish diaspora of northern Syria. Enoch is a controversial work in all its elements, not only on the above topics, in terms of calendar and myth. The figure of Enoch itself is subversive. His life story may have been inspired by the tradition of celestial travels of Mesopotamian heroes (Borger 1974; Annus 2016; Sanders 2017). However, the Mesopotamian hero returns to Earth after learning celestial secrets. Enoch's return is only intermittent, before his final takeaway to heaven, where he will serve as an intermediary between the heavenly and earthly worlds.[52] Subversiveness is a common feature of cultures that live under the attraction of an empire or under its oppression, and is characteristic of the literature of exiles.[53] The authors, regardless of their ethnicity, want to show that their heroes are successful and thrive in any circumstance. The Ahiqar novel uses the cultural language of the Neo–Assyrian empire to depict its hero escaping danger.[54] The stories in Daniel's book depict their successful hero in a Babylonian milieu, who is always saved from the malice of those who are envious. The authors of the myths of Genesis 1–11 subversively use Mesopotamian mythological themes to express their own unique anthropological and theological views, and Ezekiel does the same in his own book.

In the Enochic collection and all those familiar with this work, the original source of knowledge is the text written on the heavenly tablet, that is, a kind of cuneiform model.[55] At the same time, however, the content of these celestial tables is not the Mesopotamian ideal calendar of 360 days that serves as the basis for astral magic, but an ideal solar calendar of 364 days born by transforming the Mesopotamian tradition. In addition, the Enochic tablets serve as a source of another kind of tradition, namely the knowledge of the "good" and "bad" periods, that is, the interpretation of human history. The historical model of the "bad" eras is represented by the Watchers and their rebellion.

The authentic transmitter of the tradition from the tablets is Enoch, who conveys all these in the Aramaic books attributed to him. It is he who occupies the place of the cuneiform scribes, the *ṭupšar Enūma Anu Enlil*, connoisseurs of celestial mysteries, in Aramaic science (Drawnel 2010). In fact, he surpasses the Mesopotamian scribes: as "the scribe of interpretation", the only person who is capable of authentically interpreting texts written on celestial tablets. In addition, Enoch knows the secrets not only of the heaven, but also of history.

Enoch appears as such in works of later Jewish tradition, some of which survive under his name (2Enoch, 3Enoch). The earliest of these is the Book of Jubilees, which is a merging of the narrative tradition of Genesis and Exodus (from creation to the giving of the law of Mount Sinai) with the Enochic tradition. In Jubilees, Enoch is sitting in heaven and recording on his tablets and books the course of human history and the punishment of sinners.[56] Unlike his Mesopotamian colleagues, Enoch never returns to Earth, but mediates between the celestial world and humans from above after being "taken away" from Earth (Gen 5:24).

**Funding:** This research received no external funding.

**Informed Consent Statement:** Not applicable.

**Data Availability Statement:** No new data were created or analyzed in this study.

**Conflicts of Interest:** The authors declare no conflict of interest.

## Notes

[1] The third wave of resettlement came in 582 BCE, when new groups of the Judaean population were resettled in Babylonia. For a standard summary of events, see (Miller and Hayes 2006).

[2] According to biblical accounts, the Assyrian ruler, to be identified with Sargon (721–705 BCE), settled them "in Halah, on the Habor, the river of Gozan, and in the cities of the Medes" (2 Kings 17:5–6; 18:11). Sargon himself reports on the event, see (Elayi 2017, pp. 51–61) (51).

[3] For more information on resettlement practices, see (Oded 1979). On Israel, see (Radner 2019).

[4] A group of cuneiform sources mentions a settlement of exiles, *āl Yahudu* "the town of Judah". For the edition of sources, see (Pearce and Wunsch 2015). For settlements in the diaspora, see (Dandamayev 1984; Jursa 2023, pp. 148–56).

[5] On biblical sources, see (Blenkinsopp 1996, pp. 149–55; 1998).

[6] On Aramaic, see (Gzella 2015, chap. 3 (The Spread of Aramaic in the Assyrian and Babylonian Empires), and chap. 4 (Official Aramaic and the Achaemenid Chancellery)).

[7] On the library, see (Robson 2019, pp. 12–23).

[8] On the spiritual impact of historical events, see (Najman 2014, pp. 1–25) (Reboot).

[9] For a detailed discussion of the transformation of the prophetic genre, see (Collins 2015).

[10] The text of the book of Ezekiel is supplemented by numerous glosses; the book also shows evidence of editorial elaboration and expansion throughout (in those passages that speak of new and a new order for the future). Chapters 40–48 on the return of the glory of God (*kābôd*) to the rebuilt sanctuary may be attributed to a later author. For Ezekiel, see (Zimmerli 1979, 1983).

[11] (Carr 2020) regularly discusses parallels with Ezekiel on Genesis themes.

[12] The departure of glory from Jerusalem is alluded to in 1Sam 4:21–22; direct references are in Ezek 8–11.

[13] The Hebrew Bible does not consider Daniel a prophet; his text is in the collection of Writings (*ketubim*). In the Septuagint, on the other hand, Daniel is the fourth great prophet. The Qumran text 4Q174 1-3 ii.3 refers to the "book of the prophet Daniel (*dny'l hnby'*)" In Matthew 24:15 Jesus refers to "Daniel the prophet".

[14] Verses 40–44 of the interpretation contain an addition to interpret the four kingdoms as four empires, beginning with the Neo-Babylonian empire. The divided empire thus represents the Hellenistic successor states of the empire of Alexander the Great, the Ptolemaid and Seleucid kingdoms, and the stone that toppled the statue represents the Maccabean revolt. The reinterpretation put the prediction into a world-historical perspective. For a detailed analysis of the sources, see (Fröhlich 1996, pp. 21–36).

[15] Historiographic interest and patterns of historical reviews in postexilic prophetic and revelatory works are discussed in (E. Stone 2011, pp. 59–89) (chap. 3: Apocalyptic Historiography).

[16] On the Mesopotamian scribal background of Daniel, see (van der Toorn 2001).

[17] On the identification of the kingdoms and the reference of the symbols to different eras, see (Koch 1980). The fourth animal was identified by (Staub 1978) with the elephant, which was also used as a war animal by the Seleucid rulers, and the image of the elephant was also on their coins.

[18] The model of the "four kingdoms" and the the idea of the eternal rule coming in place of the fourth can be found in the oldest part of the Book of Daniel, chapter 2, written in Aramaic. The pattern is repeated in chapter 7, (also in Aramaic), and is mentioned in the subsequent visions of the Hebrew part (Dan 8–12). The earliest interpretation of the vision of Dan 2 could mean successive reigns of Neo-Babylonian rulers, while the fifth "rule" (*mlkw'*) identified with the Persian Empire, cf. (Fröhlich 1996, pp. 28–30).

[19] The elements of the Danielic visions survive in the later Jewish apocalyptic, for an overview, see (Collins 1993, pp. 52–61) (5: The Genre of the Visions). On a thematic survival of certain motifs, see (Amihay 2022).

[20] For a long time, the Enochic collection was only known in a shorter Greek, and a longer Ethiopic (Ge'ez) translation, considered the complete version. The discovery of fragmentary Aramaic copies of the work among the manuscripts of the Qumran library brought a turning point in the research. The Qumran manuscripts represent a continuous tradition. The first edition of the Aramaic fragments is (Milik 1976); a revised edition is (Drawnel 2019). Edition of the Ethiopic text, in the light of the Qumran fragments is (Knibb 1982).

[21] 1En 9:1; 10:4; 10:9; 10:11, cf. 4Q202 1 iii.7–13; similarly 1QM IX.15–16.

[22] Dan 7, ". . . the beast was put to death and its body destroyed and given over to be burned with fire" (Dan 7:11). The downfall of the fourth kingdom is a constant theme repeated in both the Aramaic and Hebrew parts of the Danielic collection (Dan 7, 8, 9, 10–12).

[23] For throne visions in ancient Judaism, see (Gruenwald 1980; Rowland 2022).

[24] Based on the type of writing (Milik 1976), 140 considers the earliest manuscripts to be of Syrian origin. Linguistically, these manuscripts also represent the oldest layer, cf. (Starr 2017, p. 277.)

[25] Several additions concerning the motives and teachings of the Watchers can already be found in the Aramaic copies of the narrative of the Fallen Watchers (1En 6–11). The Aramaic fragments of the Book of Giants that are at least the same age or even

older than the manuscripts containing the story itself contain texts based on the story of the Watchers. The tradition of the story can therefore be much older than the age of the known manuscripts.

26    For a summary of Enoch's Mesopotamian connections, see (Kvanvig 2011).

27    The *gnomon* found in Qumran is supposed to be used to determine the points of solstices and equinoxes, see (Albani and Glessmer 1997).

28    Ideal calendar is not a functioning calendar scheme; it presents a model for an "ideal" universe, a blueprint which can be used to making calculations, and to compare the actual date on which a phenomenon occurs against the corresponding schematic date; see (Brack-Bernsen 2007). Phenomena that can then be adjusted to fit the actual luni-solar calendar are the solstices and equinoxes.

29    Fragments 4Q208 and 4Q209 are copies of the same work from two different periods, respectively, the end of the 3rd century BCE and the last third of the 1st century BCE. This reflects that the tradition of the calendar was continuous in the community. See (Jacobus 2022).

30    On Babylonian astral magic and divination, see (Reiner 1995; Rochberg 2010, pp. 211–22) (Continuity and Change in Omen Literature).

31    From the root *prš*, meaning "to cut off" or "to separate".

32    (Milik 1976, pp. 152–54) reconstructed nineteen of the twenty names of the Watchers' leaders as *šmyḥzh* (My Name has seen), *'r'tqp* (The earth is power), *rmṭ'l* (Burning heat of God), *kwkb'l* (Star of God), *r'm'l* (Thunder of God), *dny'l* (Judge of God), *zyqy'l* / *zyq'l* (Lightning-flash of God, *brq'l* (Lightning of God), *'s'l* (God has made), *ḥrmny* (of Hermon), *mṭr'l* (Rain of God), *'nn'l* (Cloud of God), *stw'l* (Winter of God), *šmšy'l* (Sun of God), *śhr'l* (Moon of God), *tmy'l* (Perfection of God), *ṭwry'l* (Mountain of God), *ymy'l* / *ym'l* (Day of God), *yhdy'l* (God will guide). According to the details of the teachings given to the women, each Watcher taught omen interpretations related to the phenomenon in their name. On the names, see also (Langlois 2010).

33    According to (Nickelsburg 1977) the story keeps the memory of the wars of Alexander the Great's generals, the Diadochoi; (Suter 1979, 2002) saw in it a reflection of the priestly mixed marriages of the Persian era; according to (Bhayro 2005, pp. 23–25) the figures of the Watchers were inspired by the Mesopotamian *bāru*-priests; (Annus 2010), on the other hand, considers the story a polemical narrative on the tradition of the antediluvian sages, the *apkallu*-s.

34    In biblical language, this appears in the late seventh century, when under Assyrian influence "a new idea of the sky arrived in Jerusalem", see (Halpern 2003, p. 326).

35    (Rochberg 2010, pp. 135–42) (Benefic and Malefic Planets in Babylonian Astrology). According to the concept of *melothesia*, the parts of the human body are under the benevolent and malevolent influence of the zodiacal signs; some celestial bodies can even attack humans, see, (Geller 2014, p. 92). Descriptions of diseases mention the "seed" or "sperm" of the stars (*rihût kakkabim*), see (Reiner 1995, pp. 101–2). This concept also applies to dew, which Greek magical texts and herbal books also consider "the sperm of the stars".

36    The Enochic Book of Parables (1En 37–71) says that "their movement (that is, that of the heavenly bodies) is according to the number of the angels" (1En 43:2).

37    First articulated in a seminal article by (Stone 1976).

38    On revelation as a Mesopotamian scribal construct, and its impact on Hebrew writings, see (van der Toorn 2007, pp. 205–32) (Inventing Revelation: The Scribal Construct of Holy Writ).

39    In addition to the Astronomical Book, tablets read by Enoch are mentioned in 1En 93:2; 103:2; 106:19. Enoch appears in the revelation scenes as a quasi-angelic figure, see (Stone 2015).

40    The Old Testament book of Esther mentions repeatedly the term "copy of document" (*ptšgn hktb*), speaking in a context of Persian chancellery (Esth 3:14; 4:8; 8:13). The word *iggeret/egirta* is probably a loanword from the Akkadian *egirtu*, which primarily denotes a letter containing a royal command or missile letter.

41    Further references are in 1En 75:3; 75:4 and 80:1.

42    The semiotization of history from an ethical point of view is a constant feature of Old Testament historiography and the view of history in other genres as well (prophecy). In the same vein, the historical overview of the Damascus Document from Qumran, which is familiar with the Enochic tradition of Fallen Watchers, was conceived (CD II.16-III.12). On the interpretation of history in Qumran, see (Fröhlich 2011, 2017).

43    Books are mentioned in various contexts as books of the living (1En 47:3); books transmitted to Enoch's son Metushelah (1En 82:1); sealed books opened at the judgment (1En 90:20); the false books of sinners, and the books of Enoch (1En 104:10–13); the names of the sinners will be erased from the book of life and from the books of the holy ones (1En 108:2); books and records about humans in heaven above, so that the angels may read them and know what will happen to the sinners and the spirits of the humble (1En 108:7); Enoch recounts blessings of the righteous in the books (1En 108:10).

44    Cf. 1En 81:5, "Those seven holy ones brought me and set me on the earth in front of the gate to my house. They said to me: 'Tell everything to your son Methuselah'".

45    This concept is reminiscent of the attitude towards the Babylonian *Poem of Erra and Ishum*, according to which Erra, the god of the pestilence and plagues, was unleashed, causing chaos and plague in Babylon. His anger was eventually calmed by his

vizier, Ishum, and the god Marduk was allowed to return to his city. The work is one of the few ones whose author is known. The scribe Kabti-ilani-Marduk identifies himself in the colophon and explains that the poem was revealed to him in a dream directly by Erra himself. The text issuing from the god was meant to provide protection against calamities to those who knew and possessed it. Often called "plague amulet", excerpts from the composition written on clay or stone tablets were intended to protect against harms. Enoch's teachings written down and passed down through the generations serve purposes similar to Kabti-ilani-Marduk's "chain letter".

46  On alphabetic scribes in Mesopotamia, see (Bloch 2018).

47  Another and well-known idea of the heavenly world is that of the heavenly sanctuary, in which beings similar to the members of the heavenly court, called by different names, serve. Such a world vision is presented in the collection known as The Songs of the Sabbath Sacrifice from Qumran (4Q400–407 (4QShirShabb[a–h]).

48  Some examples are mentioned in (Van De Mieroop 2023, pp. 187–88).

49  The text of the Uruk List of Kings and Scholars contains the names of scholarly scribes, several of whom are historically attested. For a new edition of the text, see (Lenzi 2008).

50  For a brief summary of the rich literature on Ahiqar, see (Bledsoe 2020).

51  (Wacholder 1990) was in its time a pioneering brief analysis of these works.

52  Cf. Gen 5:24; in the Genesis Apocryphon Lamech sends his father Methuselah to the ends of the Earth to Enoch to learn the secret of the origin of the unborn Noah (1Q20 (1QapGen ar) II.19–26). The Book of Enoch itself is said to contain Enoch's revelations from heaven. The Book of Jubilees regularly refers to Enoch as the one who monitors and keeps track of earthly events from heaven.

53  (Brueggemann 1991) showed the imperial concept of some theological concepts in the language of the book of Jeremiah. For the Mesopotamian parallels of Gen 1–11, cf. collection of articles (Hess and Tsumura 1994).

54  On the subversiveness of the Ahiqar novel, see (Van De Mieroop 2023, p. 168).

55  The impact of the Enochic tradition was fundamental to the revelatory tradition of the Dead Sea Scrolls, the ways and forms of which are discussed in (Jassen 2007).

56  Jubilees, a Jewish pseudepigraphic work, survives only in an Ethiopic (*ge'ez*) translation, along with a fragmentary Latin translation and Greek quotations. Fragments of its original Hebrew text were found in Qumran. The work, which dates to around 200 BCE, had a great influence on Qumran literature and an important role in the life of the Qumran community.

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
