# Peer review of "Scribal Revelations in Ancient Judaism"

_religions, doi:10.3390/rel15010131_

Round 1

Reviewer 1 Report

Comments and Suggestions for Authors

 The proposed article is clear, scientifically sound, relevant for the field and presented in a well-structured manner. It proves the author’s extensive knowledge of the topic and excellent familiarity with both older and newer relevant literature in the field. The conclusions are consistent with the evidence and arguments presented and provide an advancement of the current knowledge. Although the topic in itself is not new, its approach is original and innovative by looking at both the religious and the historical aspects in a comparative frame with different but connected cultures. The article could generate additional research that might further the science.

Author Response

Thank you very much for your generally positive review of my article.

Reviewer 2 Report

Comments and Suggestions for Authors

This is a fine study drawing important connections between 1 Enoch and the Book of Daniel. There are three areas that require improvement before publication:

1. There are some inconsistencies of style that should be standardized. Author uses both BC and BCE date designations. BCE / CE are better in scholarship for being neutral, but in any case one should not go back and forth between the two systems. Also per style: some subsection headlines are not marked as such, others are numbered, others yet are italicized. All of this requires standardization, perhaps by omitting some headers (the article is short enough so as not to merit more than the primary 6 headers).

2. The author should be clearer on their view regarding the relationship of biblical prophetic activity and apocalyptic writing. Reference to John Collins's "Transformation of Prophecy" (2015) and Hindy Najman's Losing the Temple (2014) could be helpful here, as well as Michael Stone's Ancient Judaism (2011; chapter 3). In regards to the Dead Sea Scrolls, consider Alex Jassen's Mediating the Divine (2007).

The analysis of Danielic visions before proceeding to Enoch is done very well, but could be fortified by showing more similarities between Daniel and other apocalyptic writings. Consult Collins's Hermeneia commentary for this purpose. As an example of what I mean, note the discussion of Daniel 2 can engage with the destructive role of the stone in 4 Baruch. See Amihay, "The Stones and the Rock" (2022).

3. Most importantly, for the topic of bookish revelations, the article completely ignores Michael Stone's seminal article "List of Revealed Things." Please read and incorporate for historical and theological context of the claims brought forth in the paper.

Thanks for an enjoyable piece!

Author Response

Thank you very much for your review, which was very useful for me. I implemented your suggestions in all three areas. 1/ BC and BCE date designations were standardised as BCE. The headlines were standardized, omitting several of them: in the end, five headlines remained, without numbering, in bold italics.2/ Thanks for the bibliographic additions, they were added to the text. Special thanks for drawing my attention to the independent survival of certain Danielic motifs in later literature.3/ Unfortunately, it was only after submitting the manuscript (which, as usual, was done in a hurry because of the deadline) that I noticed the absence of several essential works from the notes and bibliography. I would definitely have added these (Stone, List of Revealed Things). The titles you suggested have all been added (plus Stone, Enoch and the Fall of the Angels: Teaching and Status, Dead Sea Discoveries 2.3 (2015), 342-357).

Reviewer 3 Report

Comments and Suggestions for Authors

1) The term "Bookish" in the title is a wrong choice. The word means something else and cannot be used in this context. Suggested improvement: "Scribal revelations" or "scriptural revelations". 

2) One misses reference to and discussion of Karel van der Toorn's Scribal Culture and the Making of the Hebrew Bible (Harvard UP 2007); in particular the section on Daniel would profit by the use of the important study of Daniel's scribal background by van der Toorn (Scholars at the Oriental Court: The Figure of Daniel Against its Mesopotamian Background in: J.J Collins (ed.), The Book of Daniel, Volume 1 Composition and Reception, Leiden, 2000.

3) I am signalling a number of minor corrections in the file attached.

Author Response

Thank you very much for your review, which was very useful for me. I implemented your suggestions.

1/ The term “bookish” was changed to “scribal.”

2/ References to Karel van der Toorn’s works were introduced.

3/ Special thanks for your careful revision and signalling minor corrections, I implemened all but one (line 252, capitalized words remain because they are terms of Sumerian origin, traditionally transcribed with capital letters). The headlines were standardized, omitting several of them: in the end, five headlines remained, without numbering, in bold italics.